# Enhancing Multi-Step Reasoning Abilities of Language Models through Direct Q-Function Optimization

## Abstract

Reinforcement Learning (RL) plays a crucial role in aligning large language models (LLMs) with human preferences and improving their ability to perform complex tasks. However, current approaches either require significant computational resources due to the use of multiple models and extensive online sampling for training (e.g., PPO) or are framed as bandit problems (e.g., DPO, DRO), which often struggle with multi-step reasoning tasks, such as math problem-solving and complex reasoning that involve long chains of thought. To overcome these limitations, we introduce Direct Q-function Optimization (DQO), which formulates the response generation process as a Markov Decision Process (MDP) and utilizes the soft actor-critic (SAC) framework to optimize a Q-function directly parameterized by the language model. The MDP formulation of DQO offers structural advantages over bandit-based methods, enabling more effective process supervision. Experimental results on two math problem-solving datasets, GSM8K and MATH, demonstrate that DQO outperforms previous methods, establishing it as a promising offline reinforcement learning approach for aligning language models.

## 1 Introduction

Large language models (LLMs) have shown remarkable performances and potentials on a wide range of tasks including dialog generation (Han et al., 2024), general question answering (Alawwad et al., 2024), code generation (Jimenez et al., 2023; Chen et al., 2024b), agents (Wang et al., 2024) and math problem solving (Yu et al., 2024; Shao et al., 2024). To ensure good performance, one of the key procedures is to align the language models with human preferences or task-specific requirements by reinforcement learning (RL) (Bai et al., 2022; Touvron et al., 2023). Canonically, the alignment training pipeline consists of two stages (Ouyang et al., 2022). In the first stage, a reward model under the Bradley-Terry model (Bradley & Terry, 1952) is trained on human or environment-labeled preference data. Then the language model is trained by online RL algorithms like Proximal Policy Optimization (PPO) (Schulman et al., 2017) with the reward signals provided by the reward model trained in stage one.

Despite the good performance achieved, the online RL methods usually involve sampling during training, which is both costly and unstable compared to offline methods (Choshen et al., 2020). These issues are overcome by offline preference learning methods, of which the representative is Direct Preference Optimization (DPO) (Rafailov et al., 2024). DPO and its follow-ups (e.g., Zhao et al. (2023); Azar et al. (2024)) treat the language model as the policy model and reward model simultaneously and train the model on offline pairwise preference data directly, therefore eliminating the need for a separate reward model. Though simple, direct preference learning has been shown effective and efficient in LLM alignment (Tunstall et al., 2023).

However, in practice, sometimes it is hard to acquire pairwise data required by the above methods. This issue becomes particularly severe under scenarios like math problem solving or code generation when generating high-quality data requires domain-specific expertise (Saunders et al., 2022; Bowman et al., 2022). This drawback of DPO has recently been circumvented by Direct Reward Optimization (DRO) (Richemond et al., 2024). DRO formulates the LLM generation task as a single-step MDP (i.e., bandit) and adopts the framework soft actor-critic (SAC) (Haarnoja et al., 2018), where the advantage is directly parameterized by the language model. Consequently, DRO inherits the advantage of offline policy gradient and gets rid of the dependency on pairwise data.

Table 1: A comparison between DQO, offline learning algorithms, including supervised fine-tuning (SFT), reject sampling (RS) (Dong et al., 2023), DPO (Rafailov et al., 2024), KTO (Ethayarajh et al., 2024), DRO (Richemond et al., 2024) and online algorithm PPO (Schulman et al., 2017). DQO enjoys all the benefits listed in the left-most column.

| | SFT | RS | DPO | KTO | DRO | PPO | **DQO** |
|---|---|---|---|---|---|---|---|
| Free from online sampling during training | ✓ | ✓ | ✓ | ✓ | ✓ | ✗ | ✓ |
| Learn from negative samples | ✗ | ✗ | ✓ | ✓ | ✓ | ✓ | ✓ |
| Learn from unbalanced samples | ✗ | ✗ | ✗ | ✓ | ✓ | ✓ | ✓ |
| Ability to use process supervision | ✗ | ✗ | ✗ | ✗ | ✗ | ✓ | ✓ |

Nevertheless, DRO treats the process as a bandit problem, which neglects the intrinsic long-horizon nature of a wide spectrum of tasks that require complex multi-step reasoning like math problem solving and code generation (Kang et al., 2024; Miao et al., 2024), where an erroneous reasoning is almost fatal. Previous RL research found that if rewards are only provided at the end of the episode, discovering this sparse reward signal is a hard exploration problem and sparse reward functions may not be able to meaningfully distinguish between a wide range of different policies, which makes the training inefficient (Riedmiller et al., 2018; Wilcox et al., 2022). In the meanwhile, recent studies show that signals from process reward models (PRMs) can further boost the performance of language model (Zhang et al., 2024a; Lightman et al., 2023). The positional information of PRM scores usually implies the critical mistakes in the reasoning and therefore provides stronger supervision signals. However, if the whole generation process is simplified as a single action, the process rewards will be aggregated and the positional information will be lost, implying that DRO cannot efficiently utilize process supervision.

In order to overcome the aforementioned issues, in this paper, we propose Direct $Q$-function optimization (DQO), an offline RL algorithm for LLMs. In DQO, the responding procedure is formulated as a Markov Decision Process (MDP) and our goal is to learn an optimal policy under KL-regularization. Our algorithm adopts the framework of soft $Q$-learning, where the $Q$-function is directly parameterized by the language model. Then both the $Q$-function network and the value network are updated according to Soft Bellman Equation on offline data. The MDP formulation makes DQO a multi-step learning algorithm, and can therefore exploit process reward signals. A holistic comparison of our method and other alignment methods is shown in Table 1. Unlike DPO or DRO, where all tokens (actions) within a response (trajectory) are evenly incentivized or punished, DQO can discover the correct reasoning steps in a incorrect response, see the case in Table 2. Specifically, our contributions are summarized as follows

- We propose Direct $Q$-function optimization, or DQO, an offline RL algorithm for LLMs. Compared to previous methods, DQO can learn from offline and negative or unbalanced samples. Moreover, DQO is featured by step-wise learning, which is favorable for long-horizon tasks and able to exploit process rewards.

- We introduce a practical instantiation of DQO, which equips DQO with $\lambda$-return and importance sampling. These techniques stabilize the training process and ensure a good performance.

- We empirically compare DQO with a wide range of widely used alignment algorithms on math problem-solving tasks. Experiment results show that DQO outperforms these baselines on both GSM8K and MATH datasets. Moreover, as shown by our experiment, when process rewards are available, the performance of DQO can be further boosted, indicating that DQO can benefit from process rewards.

## 2 PRELIMINARIES

In this section, we introduce the foundational concepts and notations that underpin our proposed algorithm. We first review the basic framework of modeling language generation as a reinforcement learning task, followed by a KL-regularized reinforcement learning objective.

**Modeling Language Generation as Token-Level MDP** Reinforcement Learning (RL) is concerned with learning a policy that maximizes the cumulative reward for an agent interacting with an environment. In this work, we formalize language generation tasks as a Markov decision process (MDP). We denote prompt as $x$ and a response to the prompt as $y$, which can each individually be broken down into a sequence of tokens, for example, $x = (x_0, \ldots, x_m)$, from a fixed discrete vocabulary

Table 2: A case study of learned $V$-function. We select a prompt-response pair from the GSM8K training set and split the response into nine segments by double line breaks. We extract the $V$-function value at each line break. The model makes a mistake at the beginning of the fifth segment (where $\$\$x=y\$\$$ is erroneous). The model predicts high $V$-values from the first segment to the fourth segment. After that, the predictions drop significantly. The value trends preciously reflect the correctness of the reasoning and thus can serve as an informative process supervision.

| | Text | Process Value | Correctness |
|---|---|---|---|
| **Prompt** | Jorge has an equal number of baseball cards as Matias, who has 6 fewer cards than Carlos. If Carlos has 20 baseball cards, what is the total number of baseball cards the three have? 
 Please reason step by step, and produce a final answer following 4 '#', like '#### 0'. | - | - |
| **Response** | `**Step 1:  Identify the information provided**` | 0.6016 | ✓ |
| | `- Jorge has an equal number of baseball cards as Matias.`
`- Matias has 6 fewer cards than Carlos.`
`- Carlos has 20 baseball cards.` | 0.6719 | ✓ |
| | `**Step 2:  Translate the information into equations**` | 0.7969 | ✓ |
| | `Let x be the number of baseball cards Jorge and Matias each have.`
`Let y be the number of baseball cards Carlos has.` | 0.6680 | ✓ |
| | `$$x = y$$ $$y = x + 6$$ $$y = 20$$` | 0.1016 | ✗ |
| | `**Step 3:  Substitute the equations to find x**` | 0.0013 | ✗ |
| | `$$x = x + 6$$ $$0 = 6$$ This is a contradiction, so the given information is not consistent.` | 0.4688 | ✗ |
| | `**Therefore, there is no solution to the problem as the given constraints are contradictory.**` | 0.1768 | ✗ |
| | `#### 0 ⟨eos⟩` | - | ✗ |

$\mathcal{A}$. We define the token-level MDP as a tuple $\mathcal{M} = (\mathcal{S}, \mathcal{A}, \mathbb{P}, H, r, d_0, \omega)$. In the defined MDP, $\mathcal{S}$ is the space of the state consisting of all tokens generated so far, i.e., $s_t = (x_0, \ldots, x_m, y_1, \ldots, y_{t-1})$. The action space $\mathcal{A}$ is the fixed discrete vocabulary. The dynamics $\mathbb{P}$ are the deterministic transition model between tokens, i.e., $\mathbb{P}(s_{t+1}|s_t, a) = 1$ for $s_t = (x_0, \ldots, x_m, y_1, \ldots, y_{t-1})$, $a = y_t$ and $s_{t+1} = (x_0, \ldots, x_m, y_0, \ldots, y_t)$[1]. The generation process will terminate once the terminal action $\omega$ (usually end-of-sentence token) is taken or reaches the maximum horizon length $H$. The reward function $r(s, a)$ provides scalar feedback for the agent's performance after taking action $a$ in state $s$. In RLHF, the reward function is usually learned from human feedback over preferences or given by a series of rules depending on the specific tasks. The initial state distribution $d_0$ is a distribution over prompts $x$, where an initial state $s_0$ is comprised of the tokens from $x$.

**KL-Regularized Reinforcement Learning Objective** We formulate the optimization objective as a KL-regularized RL problem. Our goal is to approximate the optimal KL-regularized policy

$$\pi^* = \arg\max_\pi \mathbb{E}_{\pi, s_0 \sim d_0} \left[ \sum_{h=1}^{H} \left( r(s_h, a_h) - \beta \mathrm{KL}\big(\pi(\cdot|s_h) \| \pi_{\mathrm{ref}}(\cdot|s_h)\big) \right) \right], \quad (1)$$

where $H$ is the total number of decision steps, $s_0$ is a prompt sampled from the dataset, $r(s_h, a_h)$ is the token-level reward from the reward function, $\beta$ is the coefficient controlling the magnitude of KL-regularization and $\pi_{\mathrm{ref}}$ is the initialisation policy. In classic RLHF and most LLM-related tasks, the reward is sparse and is only applied at the terminal action $\omega$, i.e. the end-of-sentence token

---

[1]For notational simplicity, we ignore the case that LLM can call an external tool. If the tool does not introduce randomness, the state transitions are also deterministic. Even if the state transitions are random, we can use samples to approximate the state transition probabilities.

`<eos>`. However, our structure is flexible enough to incorporate both dense and sparse rewards from ruled-based reward models, turn-level reward models, process-supervised reward models (PRM), or just outcome-supervised reward models.

We consider rewriting our objective function (1) under the framework of max-entropy reinforcement learning. Specifically, we decompose the KL-regularization term $\text{KL}(\pi(\cdot|s_h)\|\pi_{\text{ref}}(\cdot|s_h))$ into cross-entropy and entropy, leading to the following objective:

$$
\begin{aligned}
\pi^* &= \operatorname*{argmax}_{\pi} \mathbb{E}_{\pi,s_0\sim d_0}\left[\sum_{h=1}^{H}\left(r(s_h,a_h)+\beta\log\pi_{\text{ref}}(a_h|s_h)+\beta\mathcal{H}(\pi(\cdot|s_h)))\right)\right] \\
&= \operatorname*{argmax}_{\pi} \mathbb{E}_{\pi,s_0\sim d_0}\left[\sum_{h=1}^{H}\left(\overline{r}(s_h,a_h)+\beta\mathcal{H}(\pi(\cdot|s_h)))\right)\right],
\end{aligned}
\tag{2}
$$

where $\mathcal{H}(\pi(\cdot|s_h)) = -\mathbb{E}_{a\sim\pi}\log\pi(a|s_h)$ denotes the entropy of the policy at state $s_h$ and the KL-regularized reward $\overline{r}$ is defined as $\overline{r}(s_h,a_h) = \beta\log\pi_{\text{ref}}(a_h|s_h) + r(s_h,a_h)$. Equation 2 leads to a maximum entropy reinforcement learning problem, which enjoys the well-known closed-form solution (Haarnoja et al., 2018) as follows:

$$
\pi^*(a|s_h) = \exp\left(\frac{Q^*(s_h,a)-V^*(s_h)}{\beta}\right),
\tag{3}
$$

where $Q^*$ and $V^*$ are shorthands for the soft $Q$-function $Q^{\pi^*}$ and soft $V$-function $V^{\pi^*}$. Specifically, the soft $Q$-function and soft $V$-function of arbitrary policy $\pi$ are defined as follows

$$
Q^\pi(s_h,a) = \overline{r}(s_h,a) + \mathbb{E}_\pi\left[\sum_{t=h+1}^{H}\left(\overline{r}(s_t,a_t)+\beta\mathcal{H}(\pi(\cdot|s_t)))\right)\right],
\tag{4}
$$

$$
V^\pi(s_h) = \mathbb{E}_\pi\left[\sum_{t=h}^{H}\left(\overline{r}(s_t,a_t)+\beta\mathcal{H}(\pi(\cdot|s_t)))\right)\right].
\tag{5}
$$

Equation (3) reveals that the optimal policy $\pi^*$, soft $Q$-function $Q^*$, and soft V-function $V^*$ are interdependent, which means that knowing any two of them allows us to compute the third one.

## 3 DIRECT $Q$-FUNCTION OPTIMIZATION (DQO)

### 3.1 THE DQO OBJECTIVE

We adopt the Soft Actor-Critic (SAC) learning framework to learn the state value function $V$ and state-action value function $Q$. In SAC, the $Q$-function and $V$-function, which are parameterized by $\theta$ and $\phi$ respectively, are updated by minimizing the following squared residuals:

$$
L_V(\phi) = \mathbb{E}_{s_h\sim\mathcal{D}}\left[\left(V_\phi(s_h)-\mathbb{E}_{a\sim\pi_\theta(\cdot|s_h)}\left[Q_\theta(s_h,a)-\beta\log\pi_\theta(a|s_h)\right]\right)^2\right],
\tag{6}
$$

$$
L_Q(\theta) = \mathbb{E}_{(s_h,a_h,s_{h+1})\sim\mathcal{D}}\left[\left(Q_\theta(s_h,a_h)-\overline{r}(s_h,a_h)-V_\phi(s_{h+1})\right)^2\right],
\tag{7}
$$

where $\mathcal{D}$ is the distribution of previously sampled states and actions and $\theta$ is the parameter of $Q$-function, and, essentially the policy (i.e., the LLM) in DQO. For simplicity of notations, we always set $V_\phi(s_{H+1}) = 0$ for all $\phi$ and $s_{H+1}$. As shown in (3), the optimal policy $\pi^*$, optimal $Q$-function $Q^*$, and optimal value function $V^*$ are tightly interconnected. Specifically, they satisfy the relationship $Q^*(s_h,a_h) = \beta\log\pi^*(a_h|s_h) + V^*(s_h)$. Inspired by this, we parameterize the $Q$-value-network with the policy as follows

$$
Q_\theta(s_h,a_h) = \beta\log\pi_\theta(a_h|s_h) + V_\phi(s_h),
\tag{8}
$$

where $\pi_\theta(\cdot|\cdot)$ is the policy network, or the language model in this paper. In equation (8), instead of using an additional model to parameterize $Q$-value and learning the policy from the optimal $Q$-function $Q^*$, we directly infer the policy from the $Q$-function by parameterizing it with $\pi$. Therefore, we name our algorithm as Direct $Q$-function optimization (DQO).

By plugging in equation (8) to equation (7), we can rewrite the loss function for the policy as:

$$
L_\pi(\theta) = \mathbb{E}_{(s_h,a_h,s_{h+1})\sim\mathcal{D}}\left[\left(V_\phi(s_h)+\beta\log\pi_\theta(a_h|s_h)-\overline{r}(s_h,a_h)-V_\phi(s_{h+1})\right)^2\right].
\tag{9}
$$

Substituting $\overline{r}(s_h, a_h)$ by its definition $\beta \log \pi_{\text{ref}}(a_h|s_h) + r(s_h, a_h)$, we obtain the objective function of the policy network as follows:

$$L_\pi(\theta) = \mathbb{E}_{(s_h, a_h, s_{h+1}) \sim \mathcal{D}} \left[ \left( \beta \log \frac{\pi_\theta(a_h|s_h)}{\pi_{\text{ref}}(a_h|s_h)} - \left( r(s_h, a_h) + V_\phi(s_{h+1}) - V_\phi(s_h) \right) \right)^2 \right]. \quad (10)$$

Now we come to the training loss of $V$-function (6). The following proposition shows that we can move the expectation in (6) out of the square.

**Proposition 3.1.** Let $\widetilde{L}_V(\phi)$ be defined as below, then $\widetilde{L}_V(\phi)$ and $L_V(\phi)$ defined in (6) has same gradient with respect to $\phi$.

$$\widetilde{L}_V(\phi) = \mathbb{E}_{(s_h, a_h) \sim \mathcal{D}} \left[ \left( V_\phi(s_h) - \left[ Q_\theta(s_h, a_h) - \beta \log \pi_\theta(a_h|s_h) \right] \right)^2 \right]. \quad (11)$$

Proposition 3.1 shows that we can use $\widetilde{L}_V(\phi)$ in (11) to substitute $L_V(\phi)$ in (6). To eliminate the $Q_\theta(s_h, a_h)$ in (11), we consider the soft Bellman equation:

$$Q^\pi(s_h, a_h) = \overline{r}_h(s_h, a_h) + \mathbb{E}_{s_{h+1} \sim \mathbb{P}(\cdot|s_h, a_h)}[V^\pi(s_{h+1})],$$

where under deterministic transition $s_{h+1} = \text{concat}(s_h, a_h)$ and $\mathbb{E}_{s_{h+1} \sim \mathbb{P}(\cdot|s_h, a_h)}[V^\pi(s_{h+1})]$ can be estimated by the samples in the dataset $V_\phi(s_{h+1})$. Consequently, we substitute $Q_\theta(s_h, a_h)$ in (11) by $\overline{r}(s_h, a_h) + V_\phi(s_{h+1})$ and obtain the loss for the value function $V_\phi$ in the following form:

$$\widetilde{L}_V(\phi) = \mathbb{E}_{(s_h, a_h, s_{h+1}) \sim \mathcal{D}} \left[ \left( V_\phi(s_h) - \overline{r}(s_h, a_h) - V_\phi(s_{h+1}) + \beta \log \pi_\theta(a_h|s_h) \right)^2 \right]$$

$$= \mathbb{E}_{(s_h, a_h, s_{h+1}) \sim \mathcal{D}} \left[ \left( V_\phi(s_h) + \beta \log \frac{\pi_\theta(a_h|s_h)}{\pi_{\text{ref}}(a_h|s_h)} - V_\phi(s_{h+1}) - r(s_h, a_h) \right)^2 \right]. \quad (12)$$

We notice here that the left-hand-side of (12) and (10) are the same. Consequently, we define

$$L(\phi, \theta) = \mathbb{E}_{(s_h, a_h, s_{h+1}) \sim \mathcal{D}} \left[ \left( \beta \log \frac{\pi_\theta(a_h|s_h)}{\pi_{\text{ref}}(a_h|s_h)} + V_\phi(s_h) - V_\phi(s_{h+1}) - r(s_h, a_h) \right)^2 \right]. \quad (13)$$

When $\mathcal{D}$ is composed of pre-generated offline data, we employ importance sampling to reweight the offline data, ensuring that the offline dataset can be used effectively. We defer the detailed discussion to Section 3.3. It is worth highlighting that in our formulation of DQO, we consider generating each single token as an action. If we consider generating the whole utterance as a single action and set the horizon length $H = 1$, then equation (10) and equation (12) degenerates to the loss used by DRO (Richemond et al., 2024). This means that DRO can be viewed as a special case of the learning framework of DQO.

### 3.2 MITIGATING BIAS WITH $\lambda$-RETURN

One-step temporal difference (TD) errors have high bias and perform poorly when the value function is not well-initialized, resulting in inefficient learning. To address this, we incorporate $\lambda$-return (Schulman et al., 2015) to improve the updates for $Q$-function and $V$-function. By definition, we know that $V^\pi(s_h)$ is the sum of reward gained by next $n$ actions and $V^\pi(s_{h+n})$, or formally,

$$V^\pi(s_h) = \mathbb{E}_\pi \left[ V^\pi(s_{h+n}) + \sum_{l=0}^{n-1} \left( \beta \log \frac{\pi_{\text{ref}}(a_{h+l}|s_{h+l})}{\pi(a_{h+l}|s_{h+l})} + r(s_{h+l}, a_{h+l}) \right) \right].$$

Given a trajectory $\{s_0, a_0, \cdots, s_H, a_H\}$, we use the empirical samples to estimate the $n$-step return and define the empirical $n$-step return as:

$$G_{\phi,\theta}^{(n)}(s_h) = V_\phi(s_{h+n}) + \sum_{l=0}^{n-1} \left( \beta \log \frac{\pi_{\text{ref}}(a_{h+l}|s_{h+l})}{\pi_\theta(a_{h+l}|s_{h+l})} + r(s_{h+l}, a_{h+l}) \right).$$

It is worth noticing that $G_{\phi,\theta}^{(1)}(s_h)$ is exactly the target in (12). Now we are able to define $\lambda$-return, which is the weighted average of all $n$-step returns:

$$G_{\phi,\theta}^\lambda(s_h) = \begin{cases} (1-\lambda) \sum_{n=1}^{H-h} \lambda^{n-1} G_{\phi,\theta}^{(n)}(s_h), & \text{if } \lambda < 1 \\ G_{\phi,\theta}^{(H-h)}(s_h), & \text{if } \lambda = 1 \end{cases}.$$

We replace the target for value updates in (10) from one-step return $G^{(1)}_{\phi,\theta}(s_h)$ to $\lambda$-return $G^\lambda_{\bar{\phi},\bar{\theta}}(s_h)$, where $\bar{\phi}$ and $\bar{\theta}$ is the copy of $\phi$ and $\theta$ but are not counted into the back-propagation gradients. Now we have the loss function for the value network as follows:

$$L_V(\phi) = \mathbb{E}_{s_h \in \mathcal{D}}\left[\left(G^\lambda_{\bar{\phi},\bar{\theta}}(s_h) - V_\phi(s_h)\right)^2\right]. \tag{14}$$

Similarly, we also use the $\lambda$-return $G^\lambda_{\bar{\phi},\bar{\theta}}(s_{h+1})$ to substitute the target $V_\phi(s_{h+1})$ in (10). The new loss for $Q$-function (which is parameterized by $\pi_\theta$) with $\lambda$-return is:

$$L_Q(\theta) = \mathbb{E}_{(s_h,a_h,s_{h+1})\sim\mathcal{D}}\left[\left(\beta\log\frac{\pi_\theta(a_h|s_h)}{\pi_{\text{ref}}(a_h|s_h)} - \left(r(s_h,a_h) + G^\lambda_{\bar{\phi},\bar{\theta}}(s_{h+1}) - V_\phi(s_h)\right)\right)^2\right]. \tag{15}$$

### 3.3 Reweighting Offline Data with Importance Sampling

As we have mentioned before, our targets (10) and (12) only holds when the dataset is online sampled from $\pi_\theta$. However, in this work, we focus on offline setting and the data is pre-collected from $\pi_{\text{ref}}$. Therefore, there is a **distributional shift** between the behavior policy $\pi_{\text{ref}}$ which generated the data, and the target policy $\pi_\theta$. In order to mitigate this mismatch, offline RL algorithms often incorporate regularization techniques, including constraints or regularization on the learned policy and involving importance sampling. In this work, in addition to the KL-regularized RL objective, we employ importance sampling to reweight the offline data to match the distribution of trajectories generated by the current policy.

Let $\mu$ represent the behavior policy under which the offline data $\mathcal{D}$ was generated and $\pi$ be the current online policy. When the transition $\mathbb{P}$ is deterministic, the probability of a trajectory $\tau$ under $\mu$ and $\pi$ are computed as follows:

$$\mu(\tau|s_1) = \prod_{h=1}^H \mu(a_h|s_h), \quad \pi(\tau|s_1) = \prod_{h=1}^H \pi(a_h|s_h).$$

Therefore, we know that when the offline dataset $\mathcal{D}$ is sampled from $\tau$, we have

$$\mathbb{E}_{\tau\sim\pi}[f(\tau)] = \mathbb{E}_{\tau\sim\mathcal{D}}\left[\frac{\pi(\tau|s_1)}{\mu(\tau|s_1)}f(\tau)\right] = \mathbb{E}_{\tau\sim\mathcal{D}}\left[\frac{\prod_{h=1}^H \pi(a_h|s_h)}{\prod_{h=1}^H \mu(a_h|s_h)}f(\tau)\right],$$

where $f(\tau)$ is any function of trajectory $\tau$. This indicates that we can use the importance ratio $\pi(\tau|s_h)/\mu(\tau|s_h)$ to adjust the loss. Empirically, we truncate the importance sampling rate to avoid gradient explosion caused by extreme values. The final ratio we apply is shown as follows

$$w(\tau) = \min\left(e, \prod_{h=1}^H \frac{\pi(a_h|s_h)}{\mu(a_h|s_h)}\right) = \exp\left(\min\left(1, \sum_{h=1}^H \log\frac{\pi(a_h|s_h)}{\mu(a_h|s_h)}\right)\right), \tag{16}$$

Now let $\mu = \pi_{\text{ref}}$ and $\pi = \pi_\theta$, we plug in the importance ratio $w(\tau)$ in (16) to the loss functions (14) and (15) and then obtain our final loss functions for offline learning.

$$L_V(\phi) = \mathbb{E}_{\tau\sim\mathcal{D}}\left[w(\tau)\cdot\sum_{h=1}^H\left(G^\lambda_{\bar{\phi},\bar{\theta}}(s_h) - V_\phi(s_h)\right)^2\right],$$

$$L_\pi(\theta) = \mathbb{E}_{\tau\sim\mathcal{D}}\left[w(\tau)\cdot\sum_{h=1}^H\left(\beta\log\frac{\pi_\theta(a_h|s_h)}{\pi_{\text{ref}}(a_h|s_h)} - \left(r(s_h,a_h) + G^\lambda_{\bar{\phi},\bar{\theta}}(s_{h+1}) - V_\phi(s_h)\right)\right)^2\right].$$

When computing the gradient of the loss, the importance sampling weight $w(\tau)$ is not involved in the gradient computation. The introduction of importance ration enables us to leverage offline datasets in an online RL framework, ensuring that the updated policy remains consistent with the distribution of trajectories it would encounter during online interaction.

## 4 Experiments

In this section, we conduct extensive experiments to demonstrate the effectiveness of our proposed method. Moreover, we show that our method can be further augmented by utilizing process rewards.

Table 3: Experiment results for *Qwen2-7B-Instruct* model. We use **bold** for the best and underline for the second best. DQO significantly improves the base model's performance. This improvement surpass all the baselines when doing greedy decoding. As for sampling, DQO is comparable to DPO and surpass all other baselines.

| Data | GSM8K | | MATH | |
| Generation Method | Greedy | Sample | Greedy | Sample |
|---|---|---|---|---|
| *Qwen2-7B-Instruct* | 72.77 | $60.77 \pm 1.62$ | 37.44 | $35.79 \pm 0.50$ |
| SFT | 85.06 | $84.06 \pm 0.66$ | 44.38 | $37.34 \pm 0.41$ |
| Reject Sampling | 84.15 | $84.43 \pm 0.59$ | 49.82 | $48.19 \pm 0.43$ |
| DPO | 85.35 | $\mathbf{85.67 \pm 1.01}$ | 49.36 | $\underline{48.24 \pm 0.29}$ |
| KTO | 86.35 | $83.52 \pm 0.64$ | 50.32 | $46.52 \pm 0.48$ |
| DRO | 86.73 | $82.56 \pm 0.48$ | 51.84 | $47.39 \pm 0.28$ |
| DQO | **87.95** | $\underline{85.13 \pm 0.47}$ | **51.96** | $\mathbf{49.36 \pm 0.25}$ |

Table 4: Experiment results for *Gemma-1.1-7B-it* model. We use **bold** for the best performance and underline for the second best performance. DQO significantly improves the base model's performance. On GSM8K, DQO surpasses all other baselines by a significant margin. On MATH dataset, DQO achieves a comparable performance with DRO when doing greedy decoding and outperforms all the baseline when doing sampling at inference.

| Data | GSM8K | | MATH | |
| Generation Method | Greedy | Sample | Greedy | Sample |
|---|---|---|---|---|
| *Gemma-1.1-7B-it* | 39.65 | $37.89 \pm 1.02$ | 17.04 | $16.14 \pm 0.21$ |
| SFT | 53.45 | $46.14 \pm 1.07$ | 21.64 | $18.84 \pm 0.47$ |
| Reject Sampling | 53.60 | $53.17 \pm 0.94$ | 21.74 | $20.77 \pm 0.26$ |
| DPO | 63.46 | $62.76 \pm 0.48$ | 23.18 | $23.44 \pm 0.30$ |
| KTO | 50.49 | $49.29 \pm 0.74$ | 18.56 | $18.58 \pm 0.17$ |
| DRO | 62.92 | $\underline{63.00 \pm 0.92}$ | 24.56 | $\underline{24.10 \pm 0.37}$ |
| DQO | **64.51** | $\mathbf{64.00 \pm 0.37}$ | **24.90** | $\mathbf{24.84 \pm 0.29}$ |

## 4.1 DATASETS

We evaluate the models using two widely established mathematical problem-solving datasets: MATH (Hendrycks et al., 2021) and GSM8K (Cobbe et al., 2021). The MATH dataset consists of 5,000 challenging problems in its test set and 7500 in its train set. The problems cover various fields like algebra, geometry, probability, and calculus. The GSM8K dataset consists of a train set of 7473 problems and a test set of 1319 problems. The problems are mostly simpler grade-school math word problems. The problems in both datasets usually require multi-step reasoning and complex arithmetic to derive the correct answers.

We use the 7.5K training problems from the MATH dataset and 7.47K training problems from the GSM8K dataset to generate the training corpus of our baselines and DQO. We use our base model and sample 20 responses for each prompt in the training set and then label all these responses as positive and negative responses. The detailed usage of these samples is discussed in the next section and please refer to the Appendix A for a more detailed discussion of our dataset construction.

## 4.2 MODELS, BASELINES AND EVALUATION

In our experiments, we select two pretrained models, Gemma-1.1-it-7B[2] (Gemma) (Team et al., 2024) and Qwen2-7B-Instruct[3] (Qwen) (Yang et al., 2024) as our base model. We implement our method based on HybridFlow (Sheng et al., 2024). We select SFT, reject sampling (RS), DPO, KTO and DRO as our baselines. We defer the detailed training set construction and training hyperparameters as well as other training details of the baselines and our method to Appendix B.

We evaluate the models generated by our method and baseline methods on the test set of GSM8K and MATH. We consider two different decoding strategies, greedy decoding and sampling. We set

---

[2]https://huggingface.co/google/gemma-1.1-7b-it

[3]https://huggingface.co/Qwen/Qwen2-7B-Instruct

the sampling parameters to the same as when we generated the training corpus. For each prompt in the dataset, we sampled with 5 different seeds and report the mean and standard deviation of the performance.

### 4.3 EMPIRICAL RESULTS AND ABLATION STUDIES

Here we show our main results on both the GSM8K and MATH datasets. The results are shown in Table 3 and Table 4 respectively. From Table 3, we see that all the methods improve the performance of the base models by a significant margin. Particularly, on GSM8K, DQO improves the performance from 72.77% to 87.95% for greedy generation and 60.77% to 85.13% for sampling. This improvement is comparable with DPO and surpasses DRO and other baselines by a margin of 0.70% for sampling and 1.22% for greedy generation. On MATH, we also see a significant performance improvement from DQO. As for greedy decoding, the performance of DQO, while comparable with DPO and DRO, surpasses all other baselines by a margin of 1.64%. As for sampling, DQO reaches a performance of 49.36%, which surpasses the performance of the best baseline method DPO by a margin of 1.12%. These results indicate that DQO achieves a comparable performance of DPO and surpasses other baselines by a considerable margin.

When it turns to the results on Gemma, we see that DQO enjoys larger advantages. As demonstrated in Table 4, we see that all considered methods result in significant improvement. Specifically, on GSM8K, DQO improves the base model's performance by a margin of 24.86% for greedy decoding and 26.11% for sampling. These results surpass the improvement obtained by DPO by margins of 1.05% as for greedy decoding and is comparable to the performance of DRO when sampling. The advantage is even more compared to other baseline methods. On the MATH dataset, we see that DQO also improves the model's performance by a prominent margin of 7.86% and 8.70% for greedy generation and sampling, respectively. This improvement slightly surpasses DRO and surpasses other baseline methods by a margin of at least 2.28% for greedy decoding and 2.56% for sampling. In summary, DQO results in promising improvement over the base models under all the scenarios and outperforms all our baseline methods.

#### 4.3.1 IMPORTANCE SAMPLING

Table 5: The impact of importance sampling rate on both $Q$-function loss and $V$-function loss. The experiments are conducted on Gemma. When training without an importance sampling ratio on $Q$-function loss, the performances degenerate significantly on both GSM8K and MATH. When keeping the importance ratio only on $Q$-function loss, there is also a moderate performance loss on MATH.

| Data Generation Method | | GSM8K | | MATH | |
|---|---|---|---|---|---|
| | | Greedy | Sample | Greedy | Sample |
| $Q$-loss w/o IS | $V$-loss w/o IS | 58.68 | 60.20 | 21.96 | 22.68 |
| | $V$-loss w/ IS | 56.03 | 56.48 | 20.82 | 20.94 |
| $Q$-loss w/ IS | $V$-loss w/o IS | 63.53 | **64.06** | 22.28 | 23.18 |
| | $V$-loss w/ IS | **64.51** | 64.00 | **24.90** | **24.84** |

Table 6: The impact of $\lambda$-return on Gemma. When decreasing $\lambda$ from 1.0 to 0.95, we observe a significant performance dropping more than 4.31% on GSM8K and 2.30% on MATH.

| Data Generation Method | GSM8K | | MATH | |
|---|---|---|---|---|
| | Greedy | Sample | Greedy | Sample |
| $\lambda = 0.95$ | 60.20 | 59.21 | 22.60 | 22.26 |
| $\lambda = 1.0$ | **64.51** | **64.00** | **24.90** | **24.84** |

To demonstrate the impact of the importance sampling ratio in DQO, we train DQO on Gemma without the importance sampling ratio for $Q$-function loss, $V$-function loss, and both. We present the results in Table 5. The results show that, without adding importance sampling, the performance will be significantly deteriorated. Specifically, on the GSM8K dataset, when importance sampling is not introduced to $Q$-function loss, the performance degenerates by a margin over 3.80%. Similarly, on the MATH dataset, we see that when we exclude the importance sampling ratio from $Q$-function loss, the performance decreases by a margin over 2.16%. When we keep the importance sampling ratio only on $Q$-function loss, the performance on GSM8K almost maintains but we still see a moderate performance loss on MATH. These results show that the importance sampling ratio, on both $Q$-function and $V$-function loss, plays important roles in DQO training.

### 4.3.2 λ-RETURN

In order to demonstrate the impact of $\lambda$-return, we vary the value of $\lambda$ and evaluate the training results on Gemma. Empirically, we find that the best performance is obtained at $\lambda = 1$ and quickly degenerates when decreasing $\lambda$. Therefore we pick $\lambda = 0.95$ to make the comparison. The results are shown in Table 6. When switching $\lambda$ to 0.95, we observe that the performance on GSM8K decreases by a margin of more than 4.31% for greedy generation and almost 5% for sampling. The results on MATH demonstrate a similar pattern and the performances of $\lambda = 0.95$ dropped by a margin of 2.30% on both inference strategies. The results indicate that $\lambda$-return is the key component in the target of policy training.

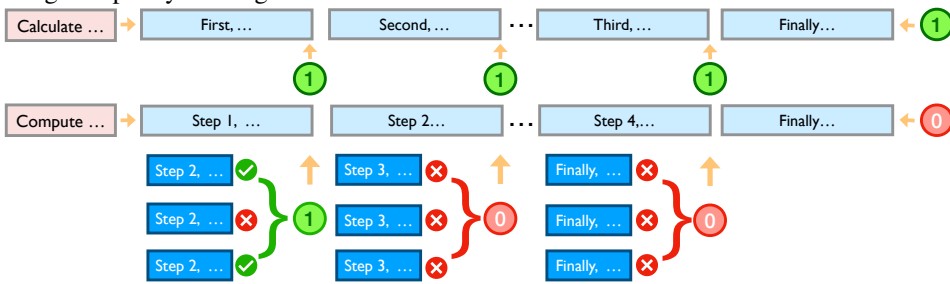

Figure 1: A visual demonstration of our process reward construction. We split all the responses to segments. For correct responses we assign all process reward to one. For negative responses, we start from each prefix and generate 20 samples. We then find the longest prefix where the best of 20 samples is correct and assign all the process rewards before to 1.

### 4.4 DQO WITH PROCESS SCORE

In this section, we show that when process scores at intermediate steps are available, the performance of DQO can be further improved. Here, we use synthetic process scores. In order to obtain a synthetic process score, we consider using an empirical passing rate to estimate the quality of a given response prefix. Specifically, given a prompt string $x$, for each failed response $y$, we first split the response into several segments $y[0:n]$, where $n$ is the number of segments and we use $y[0:i]$ to denote the concatenation of first $i$ segments. Beginning from $i = n-1$, we randomly sample 20 trajectories given prefix $\text{contat}(x, y[0:i])$. If there is at least one correct completion, we assume that the reasoning process in $y[0:i]$ is correct and all the process rewards for the previous step will be set to $1/n$. We combine these process reward scores with the original rewards. The process is summarized in Figure 1.

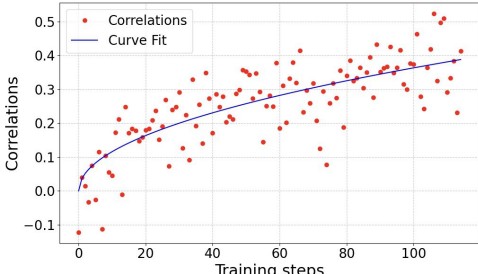

Figure 2: The correlation between the trained values (trained without process scores) and our constructed process scores. As the training proceeds, we observe a growth in the correlation between these two values, indicating that the value model in DQO learns the correctness of the reasoning process.

Table 7: Experiment results for DQO augmented by process scores. With process rewards, when using greedy decoding, the performance of DQO is further enhanced by 0.53% on GSM8K and 0.32% on MATH. The performance when doing sampling also increase on GSM8K and maintains almost the same on MATH.

| Method | GSM8K | | MATH | |
| Generation Method | Greedy | Sample | Greedy | Sample |
|---|---|---|---|---|
| *Gemma-1.1-7b-it* | 39.65 | $37.89 \pm 1.02$ | 17.04 | $16.14 \pm 0.21$ |
| DQO | 64.51 | $64.00 \pm 0.37$ | 24.90 | $\mathbf{24.84 \pm 0.29}$ |
| DQO w/ process scores | **65.04** | $\mathbf{64.55 \pm 0.66}$ | **25.22** | $24.70 \pm 0.38$ |

We conduct the experiments on Gemma, and the results are summarized in Table 7. Equipped with our estimated process scores, we see a further improvement. Specifically, on GSM8K, using our process scores further increases the performance by 0.53% for greedy decoding and 0.55% for sampling. On MATH, process scores also boost the model's performance by a further 0.32% when doing greedy decoding and maintains almost the same to DQO without process reward for sampling.

The results imply that DQO can be further enhanced by utilizing process scores. Additionally, we discovered that even without process scores, DQO is capable of identifying correct reasoning steps. The trained values demonstrate a growing correlation with the synthetic process scores, see in Figure 2, suggesting that DQO effectively learns to recognize correct reasoning steps, enhancing the reasoning process over time.

## 5 RELATED WORK

**Reinforcement Learning for Language Model Alignment** Aligning language models with human preferences, or reinforcement learning with human feedback (RLHF), dates back to the work of Wirth et al. (2017) and Christiano et al. (2017). It has been widely applied to a bunch of recent models including GPT-4 (Achiam et al., 2023), Gemini (Team et al., 2023), and Llama (Touvron et al., 2023), leading to the surprising performance of these models. The alignment procedure usually takes place after supervised finetuning (SFT). In the canonical approaches of RLHF (Ouyang et al., 2022; Bai et al., 2022; Munos et al., 2024), a reward model is first trained with preference data and then the model is updated with Proximal Policy Optimization (PPO). Another line of works, initiating from Direct Preference Optimization (DPO) (Rafailov et al., 2024), include SLiC (Zhao et al., 2023), IPO (Azar et al., 2024), KTO (Ethayarajh et al., 2024) and so on. These approaches are featured by directly parameterizing the reward models with the language model and then training on offline preference data. Following DPO, one branch of works, including GSHF (Xiong et al., 2024a), SPPO (Wu et al., 2024) and INPO (Zhang et al., 2024b), adapts DPO or its variant to online samples and iterative training and resulted to state-of-the-art models. On the other hand, Richemond et al. (2024) adapted offline reinforcement learning algorithm to direct preference learning and proposed Direct Reward Optimization (DRO), which combined offline policy learning with a value function learning and updated policy network and value network iteratively. Our work has a similar structure to DRO, but models the language generation as an MDP rather than a bandit and can utilize process supervision to facilitate training.

**Multi-step and Long Horizon RL for LLM alignment** Many tasks for LLMs require LLMs to reason step by step or interact with the environment turn by turn. However, the rewards are usually sparse since they are only provided at the end of a long horizon of reasoning or interactions. In traditional RL literature, one approach towards breaking the curse of lone horizon and sparse reward is to train or estimate an intermediate value function or process reward (Park et al., 2024) and use the process reward to guided searching (Torne et al., 2023; Zhang et al., 2024a) and RL training. The utilization of process reward has also led to better performance for LLM reasoning (Zhang et al., 2024a; Lightman et al., 2023). Most straightforwardly, Snell et al. (2023) proposed ILQL, which employed implicit Q-learning to train a Q-function network and V-function network. Then, at inference time, ILQL uses learned value functions to perturb the log probabilities of the initial policy towards utility-maximizing behavior. The success of direct preference learning also stimulates a lot of work adapting DPO to multi-turn scenarios. To estimate process reward and utilize the information provided, Chen et al. (2024a); Lai et al. (2024); Xie et al. (2024) leverages process reward signals or AI feedback to construct preference pairs for intermediate steps and update the model with original DPO. On the other hand, Xiong et al. (2024b); Shani et al. (2024) extends the vanilla DPO to accommodate the multi-turn structure. However, these approaches require pairwise data, which might not be available or easy to obtain on some specific occasions. Our work, while following the approach of direct preference learning, eliminates the need for pairwise data and can be boosted by process rewards.

## 6 CONCLUSION

In this work, we propose DQO, an offline reinforcement learning algorithm for enhancing the language model's ability in multi-step reasoning. Compared to previous online methods like PPO, the offline nature of DQO bypasses the requirement of an extra reward model and online sampling during training. Previous offline methods usually formulate the LLMs' responding process as a bandit problem, which usually fails to capture the implicit long-horizon and multi-step nature of those tasks requiring a long chain of thought. In contrast, DQO frames the tasks as a Markov decision process and further employs a soft actor-critic framework to learn the $V$-function and the $Q$-function, which is directly parameterized by the language model. To verify the effectiveness of DQO, we conduct extensive experiments on two math-problem-solving datasets, GSM8K and MATH, and empirical results show that DQO outperforms all our baseline. Currently, our experiment results are limited to two base models due to time constraints and we leave it as future work.

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

## A  DATASET CONSTRUCTION

Table 8: The distribution of positive and negative trajectories sampled from the base models. These data are directly applied to train DQO, DRO and KTO.

| Model | Qwen | | Gemma | |
|---|---|---|---|---|
| Dataset | GSM8K | MATH | GSM8K | MATH |
| # Positive Responses | 104105 | 71776 | 57922 | 22600 |
| # Negative Responses | 45355 | 78224 | 91538 | 127400 |

In this section, we provide details about the generating process of our training data. For all problems, we add format guide to make to language models' generation follows the same format as the solution provided by the dataset. Specifically, we use the following templates for GSM8K and MATH respectively

- GSM8K: {{ problem }} \n Please reason step by step, and produce a final answer following 4 '#', like '#### 0'.

- MATH: {{ problem }} \n Please reason step by step, and put your final answer within \boxed{}.

The prompt is further wrapped by chat-ML template. We then sample 20 responses for each prompts with sampling parameter `top_p`=0.9, `top_k`=16 and `threshold`=0.01. For each response, we use regular expression to unwrap the answer and match it with the reference to generate the label. The distribution of positive and negative responses are shown in Table 8. For reject sampling, we collect all correct response as the training target. For DPO, we first pairs up all positive and negative responses and then randomly sample from all possible pair to make the size of DPO training set approximately half of DQO, which means the dataset contains a similar number of trajectories. We summarize the dataset size for each baselines in Table 9.

Table 9: The size of datasets for all of our baselines and DQO. We ensure that the size of DPO training set is at least half of the training set of the training set for DQO. This guarantees that the number of trajectories in DPO dataset is no less than the number of trajectories in the dataset of DQO for a fair comparison.

| Model | Dataset | SFT | RS | DPO | KTO/DRO/DQO |
|---|---|---|---|---|---|
| Qwen | MATH | 7500 | 71776 | 87424 | 150000 |
| | GSM8K | 7473 | 94889 | 87996 | 149460 |
| Gemma | MATH | 7500 | 22600 | 94624 | 150000 |
| | GSM8K | 7473 | 46062 | 79523 | 149460 |

## B  ADDITIONAL EXPERIMENT DETAILS

We conducted all our experiments on $8\times$ NVIDIA A100 GPUs with approximately 80G memories. The training time is less than 1 hours for SFT, about 1 hours for RS, 4 hours for DPO, KTO, 6 hours for DRO and 10 hours for DQO. We summarize the detailed hyperparameters as follows.

**SFT and Reject Sampling**  For SFT and reject sampling, we select the best learning rate from {2e-5, 1e-5, 5e-6, 1e-6} and the best epoch from {1,2,3}. For SFT, the final learning rate is set to

2e-5 for Qwen and 5e-6 for Gemma. For reject sampling, the final learning rate is set to 2e-5 for Qwen and 1e-6 for Gemma. We set global batch size of 8 and therefore global batch size to 64. Both SFT and reject sampling are trained for 3 epochs. We trained the model for 3 epoches for both SFT and reject sampling.

**DPO and KTO**   For DPO, we tried $\beta$ from $\{0.1, 0.01\}$ and learning rate from $\{5e\text{-}7, 1e\text{-}7, 5e\text{-}8\}$ and select the hyperparameter set that yields the best performance. Specifically, for both Qwen and Gemma we set $\beta$ to 0.1 and learning rate to 5e-8. We set the local batch size to 8 and therefore global batch size to 64. We train the model for 1 epoch. As recommended by the original paper of KTO, We adapt directly transferred our hyperparameters of DPO to train KTO.

**DRO and DQO**   For both DRO and DQO, we tries the KL-regularization parameter $\beta$ from $\{0.01, 0.03, 0.1, 0.3, 1\}$ and learning rate from $\{5e\text{-}7, 1e\text{-}7, 5e\text{-}8\}$ and then select the best parameter set that yields the best results. The final parameter for both DRO and DQO is $\beta = 0.03$ and we set learning rate to 5e-7 for Qwen and 1e-7 for Gemma. We set the local batch size to 32 and therefore global batch size to 256. We train the model for a maximum of 5 epochs and select the best check-points on the training curve for evaluation. Finally, we select the DRO checkpoints after the first epoch and DQO checkpoints at the end of the second epoches.

