# OpenReview forum: "Enhancing Multi-Step Reasoning Abilities of Language Models through Direct Q-Function Optimization"
_ICLR.cc/2025/Conference — Submitted to ICLR 2025_

### Official Review · Reviewer_bDF5 · 2024-11-04

**Soundness:** 2
**Presentation:** 3
**Contribution:** 3
**Rating:** 5
**Confidence:** 4

**Summary:**

This paper introduces the Direct Q-Function Optimization (DQO) approach, which leverages a Markov Decision Process (MDP) formulation to enhance the performance of large language models (LLMs) on reasoning tasks. By adopting the multi-step nature of MDPs, DQO addresses limitations found in bandit-based offline methods, such as DRO, and remains effective even when preference-based datasets are unavailable.

**Strengths:**

1. The proposed algorithm frames response generation as a MDP, making it more suitable for long-horizon, step-by-step reasoning tasks compared to DRO.
2. The approach demonstrates potential in effectively leveraging process rewards to enhance performance through feedback at each step.
3. DQO incorporates lambda-return and importance sampling to ensure efficient use of offline data.
4. Experimental results highlight the superior performance of DQO on widely used GSM8K and MATH datasets.
5. The paper is well-written and easy to follow.

**Weaknesses:**

1. Fairness of Comparison:  According to Table 8, the size of datasets for each model differs significantly.  For example, DQO uses datasets up to four times larger than those for DPO. Although DPO's data subset is sampled from DQO's dataset, this disparity raises concerns about fair comparisons, especially given the relatively small performance margin of DQO over DPO for the Qwen2-7B-Instruct model when dataset differences are minimal.

2. Lack of Experimental Details: The paper lacks sufficient discussion on the evaluation of generated/augmented responses, the distribution of positive and negative responses in the training and testing data, and detailed training parameters (e.g., number of epochs for DRO and DQO). This makes it difficult to directly explain performance discrepancies, such as those seen between this work and [A1] for Qwen2-7B-Instruct (see Question 3).
3. Hard to assess the benefits of  MDP formulation: Without importance sampling, DQO’s performance (as shown in Table 5) is worse than DRO’s performance in Table 4. This raises the question of whether the key factor in DQO's superior performance is the MDP formulation itself or the use of importance sampling.
4. Risk of overfitting: The dependency on offline data and importance sampling introduces the risk of overfitting to this data, especially if it does not represent the diverse scenarios encountered in real-world applications.
5. Potentially high computational costs: DQO involves complex training procedures, including learning Q and V functions and using  \lambda-return and importance sampling. This complexity may lead to higher computational costs compared to DRO, especially for long-horizon tasks. The paper does not discuss computational efficiency; for example, chain-of-thought prompting [A1] can also provide intermediate checks without additional training. A more thorough comparison with CoT and [A2] is recommended.
6. Clarification of Specific Challenges: While this work effectively incorporates SAC, lambda_return, and importance sampling for improving reasoning performance, there is insufficient discussion on the challenges encountered during this integration. Highlighting these  challenges would strengthen the work.
7. Insufficient discussion on unbalanced data and process reward: While Table 1 notes that DQO can learn from unbalanced samples, no experiments substantiate this claim. In Section 4.4, a synthetic process reward mechanism is presented, but more explanation and experimentation with different process score designs are needed.

**Questions:**

1. How were generated responses evaluated for each augmented dataset? What is the distribution of positive and negative responses in the training and testing data?

2. For Qwen2-7B-Instruct, why does DPO consistently outperform KTO and DRO, while DRO outperforms DPO for Gemma-1.1-7B-it? Could the authors provide insights into this discrepancy??
3. The related work [A1] reports pass rates of 82.3% and 49.6% for Qwen2-7B-Instruct on GSM8K and MATH, respectively. Why does this differ from the results in Table 3?
4. Could the authors explain why DQO with lambda=1 outperforms lambda=0.95?
5. What is the correlation between the learned value function and constructed process scores during testing?

Minor issue:
1. Undefined/Misleading Notations: In eq (6), the expectation should be taken w.r.t. a_h. In Line 247, it should be V^{\pi}. The update equation (16) should be G^{\lambda}_{\phi, \theta}(s_h) = G^{(H-h)}_{\phi, \theta}(s_h) when \lambda=1.
1. Grammar and Typos: The Line 240-241 should reference DRO instead of DQO.



[A1] Lai, Xin, et al. "Step-dpo: Step-wise preference optimization for long-chain reasoning of llms." arXiv preprint arXiv:2406.18629 (2024).

[A2] Wei, Jason, et al. "Chain-of-thought prompting elicits reasoning in large language models." Advances in neural information processing systems 35 (2022): 24824-24837.

---

> ### Author Response · Authors · 2024-11-22
>
> **Q1**: Fairness of Comparison
>
> **A1**: We thank the reviewer for the suggestion and added more pairs to DPO training data. We ensure that under all setting the number of pairs of DPO is at least half of the numbers of trajectories for training DQO, which means that DPO is always using more trajectories than DQO. Specifically, the size of our new training set of DPO are shown as follows
>
> | Setting | Qwen + GSM8K | Qwen+ MATH | Gemma + GSM8k | Gemma + MATH |
> | ------- | ------------ | ---------- | ------------- | ------------ |
> | # Pairs | 87996        | 87424      | 94624         | 79523        |
>
> The following table summarizes the results of new results on Qwen.
>
> | Method     | Qwen  + GSM8K (greedy) | Qwen  + GSM8K (sample) | Qwen  + MATH (greedy) | Qwen  + MATH (sample) |
> | ---------- | ---------------------- | ---------------------- | --------------------- | --------------------- |
> | Base Model | 72.77                  | 60.77$\pm$ 1.62        | 37.44                 | 35.79$\pm$0.50        |
> | DPO        | 85.35                  | 85.67$\pm$1.01         | 49.36                 | 48.24$\pm$0.29        |
> | DRO        | 86.73                  | 82.56$\pm$0.48         | 51.84                 | 47.39$\pm$0.28        |
> | DQO        | 87.95                  | 85.13$\pm$0.47         | 51.96                 | 49.36$\pm$0.25        |
>
> The following table summarizes the results of new results on Gemma
>
> | Method     | Gemma  + GSM8K (greedy) | Gemma  + GSM8K (sample) | Gemma  + MATH (greedy) | Gemma  + MATH (sample) |
> | ---------- | ----------------------- | ----------------------- | ---------------------- | ---------------------- |
> | Base Model | 39.56                   | 37.89$\pm$1.02          | 17.04                  | 16.14$\pm$0.21         |
> | DPO        | 63.46                   | 62.76$\pm$0.48          | 23.18                  | 23.44$\pm$0.30         |
> | DRO        | 62.92                   | 63.00$\pm$0.92          | 24.56                  | 24.10$\pm$0.37         |
> | DQO        | 64.54                   | 64.00$\pm$0.37          | 24.90                  | 24.84$\pm$0.29         |
>
> The results are also updated in our paper. We see that DQO still outperforms DPO and other baselines.
>
> **Q2**: Lack of Experimental Details; How were generated responses evaluated for each augmented dataset? What is the distribution of positive and negative responses in the training and testing data?
>
> **A2**: We thank the reviewer for the suggestions. In fact, as shown in Appendix A, our prompt requires the model to wrap its answer after "####" for GSM8K and in "\box{}" for MATH. Then for each response we use regular experession to extract the answer and use the official scripts to decide whether the extracted answer is correct. Following the suggestion of the reviewer, since the testing data only includes problems, we report the distribution of positive and negative responses in our generated training data as follows.
>
> | Method   | Qwen + MATH | Qwen + GSM8K | Gemma  + MATH | Gemma  + GSM8K |
> | -------- | ----------- | ------------ | ------------- | -------------- |
> | Positive | 71776       | 104105       | 22600         | 57922          |
> | Negative | 78224       | 45355        | 127400        | 91538          |
>
> We have also rewrite related parts in our revision.
>
> **Q3**: Hard to assess the benefits of MDP formulation
>
> **A3**: We would like to remind the reviewer that importance sampling is a key component when applying soft-actor-critic to offline settings. Specifically, to estimate the $V$-fucntion of **current policy** $V^{\pi}(s)=E_{a\sim \pi_\theta(\cdot | s)}[Q(s,a)]$, we need to take expectation over all action $a$ with probability $\pi_\theta(a|s)$. However, since the training data is sampled from $\pi_\text{ref}$, taking expectation over $\mathcal{D}$ is essentially computing $E_{a\sim \pi_{\text{ref}}(\cdot | s))}[Q(s,a)]$, which actually equals to $V^{\pi_\text{ref}}$. Therefore, to bridge the mismatch in distribution, we have to incorporate important sampling to reweight the samples to make it align with the distribution generated from $\pi_{\theta}$. To summarize, importance sampling should be interpreted as part of the SAC algorithm rather than a stand-alone component of DQO.

---

> > ### Author Response · Authors · 2024-11-22
> >
> > **Q4**: Risk of overfitting
> >
> > **A4**: We thank the reviewer for raising this point. In this work, we only use the prompts from the training set of MATH and GSM8k to train the model and then test the model on the test set, which contains a set of prompts different from the train set. To make this point more concrete, we examine our DQO checkpoints with the method proposed in [1]. Due to the time limit, we only examined the gemma-based checkpoints and only adopted perplexity (PPL) as the atomic detection metric The results are summarized as follows
> >
> > | Dataset | Rewrite 1 | Rewrite 2 | Rewrite 3 | Original | $\delta$ |
> > | ------- | --------- | --------- | --------- | -------- | -------- |
> > | MATH    | 8.76      | 8.43      | 8.34      | 7.69     | 10.66%    |
> > | GSM8K   | 8.42      | 8.92      | 8.49      | 8.64     | -0.34%    |
> >
> > As shown by the last column of the table, the relative difference between PPL on the original dataset and the rewritten dataset is small, indicating a small risk of data leakage. As shown by Table 3 and Table 4, DQO outperforms other baselines on the test set, showing that DQO does not simply overfit the training set.
> >
> > **Q5**: Potentially high computational costs
> >
> > **A5**: We agree that introducing an intermediate value function increases the training cost. However, as we have mentioned in the 4-th paragraph in Section 1 (starting from line 67), introducing an intermediate value function is beneficial, especially in the scenario of long-horizon and sparse reward. Compared to related work [2], our approach does not require labeling of the correctness of intermediate labeling, which might be costly under some specific scenarios (e.g., labeler or strong LLM are not available). We have added related discussions in our paper.
> >
> > **Q6**: Clarification of Specific Challenges
> >
> > **A6**:We would like to reiterate the challenge we encountered and our contributions. First, as we have argued in the introduction, one challenge in the LLM reasoning task is the curse of long horizon and sparse reward. To overcome this challenge, we formulate the task as an MDP rather than a bandit and employ SAC to solve the MDP. Second, vanilla SAC involves three models, V, Q, and policy, so directly adopting SAC is inefficient in both training and inferencing. Inspired by direct preference learning, we use LLM to directly parameterize the Q-function and eliminate the extra Q-function required by the original SAC. Third, at the beginning of training the value function is not well-trained, causing instability in training. To overcome this, we introduce lambda-return to stabilize training. Finally, to adapt SAC to offline RL training, we incorporate importance sampling to reweight samples. We would like to clarify that while we indeed incorporate SAC with lambda-return and importance sampling, the key challenge comes from offline setting and noisy initialization, which motivate such incorporation.
> >
> > **Q7**: Insufficient discussion on unbalanced data and process reward
> >
> > **A7**: We thank the reviewer for recognizing these advantages of our method. Actually, our train set is constructed by rolling out the base model, which means that for some problems, the model will be correct or make mistakes for all 20 trials. Then for these prompts, the sampled trajectories are unbalanced (i.e., most of the trajectories are right / wrong). As shown by the following table, trajectories for a considerable proportion of problems are unbalanced.
> >
> > | Setting                 | Qwen + GSM8K | Qwen+ MATH | Gemma + GSM8k | Gemma + MATH |
> > | ----------------------- | ------------ | ---------- | ------------- | ------------ |
> > | # All-positive problems | 237          | 376        | 365           | 205          |
> > | # All-negative problems | 46           | 1660       | 1195          | 4363         |
> > | # Problems              | 7473         | 7500       | 7473          | 7500         |
> >
> > These data cannot be utilized by DPO but can be leveraged by DRO and DQO. Since unbalanced data makes up a large proportion of training data, the empirical success of DQO can be attributed to the ability to utilize these trajectories.
> >
> > In this paper, we mainly focus on the offline RL algorithm for LLM. Therefore, we only considered the most naive process value. As shown by Table 7, even the most naive external process value can further enhance the performance of DQO, which already shows that DQO can make use of external process values.

---

> ### Author Response · Authors · 2024-11-22
>
> **Q8**: For Qwen2-7B-Instruct, why does DPO consistently outperform KTO and DRO, while DRO outperforms DPO for Gemma-1.1-7B-it
>
> **A8**: We believe that this is because the gemma is not performing well on both datasets, therefore there might be more prompt that all trajectories cannot solve the problem and these data cannot contribute to DPO training. However, these portions of trajectories can be utilized by DRO. As a result, DRO outperforms DPO on gemma. On the other hand, the Qwen base model is performing well on both datasets, with means that there are sufficient pairs for DPO and the benefit of utilizing imbalanced data is not prominent in this case. This makes DPO outperform DRO on Qwen.
>
> **Q9**: The related work [2] reports pass rates of 82.3% and 49.6% for Qwen2-7B-Instruct on GSM8K and MATH, respectively. Why does this differ from the results in Table 3?
>
> **A9**: We believe that this might be due to different prompting strategy. As we have shown in Appendix A, we wrap the questions in the following template to prompt the LLMs.
>
> - GSM8K {{problem}} \n Please reason step by step, and produce a final answer following 4 `\#', like `\#\#\#\# 0'.
> - MATH: {{problem}} \n Please reason step by step, and put your final answer within \boxed{}.
>
> which might differ from the prompt template used in [2]. This discrepancy might introduce some discrepancy in perforamance.
>
> **Q10**: Could the authors explain why DQO with lambda=1 outperforms lambda=0.95?
>
> **A10**: We believe that this is because of the inaccurate initial estimation of V-function. Intuitively, when $\lambda=1.0$, the computed value target does not rely on the value of later states estimated by the value model, while $\lambda=0.95$ relies on such inaccurate value estimation. This introduces noisy signal to the training and causes instability, resulting to the phenomenon that $\lambda=1$ outperforms $\lambda=0.95$.
>
> **Q11**: What is the correlation between the learned value function and constructed process scores during testing?
>
> **A11**: We believe that the learned value function and constructed process scores are still fairly related on the test set. However, in our paper, the naive process score construction involves splitting each responses and sample 20 completions for each prefix of each responses, which is costly in both time and computational resources. Therefore due to time limit we are not able to report the results currently. We will report the exact results once they are available.
>
> **Q12**: Minor Issues
>
> **A12**: Thank you for pointing out these typos and we have revised them accordingly.
>
> References:
>
> [1] Xu, Ruijie, et al. "Benchmarking benchmark leakage in large language models." arXiv preprint arXiv:2404.18824 (2024).
>
> [2] Lai, Xin, et al. "Step-dpo: Step-wise preference optimization for long-chain reasoning of llms." arXiv preprint arXiv:2406.18629 (2024).

---

> > ### Author Response · Authors · 2024-11-25
> > **Looking Forward to Your Reply**
> >
> > Dear Reviewer bDf5:
> >
> > We hope this message finds you well. Thank you for your thoughtful and constructive feedback on our submission.
> >
> > We have answered all the comments and questions you raised. Specifically,
> >
> > 1. For our methodology, we illustrated the challenge in the LLM reasoning task and how we overcome these challenges by introducing DQO and incorporating it with $\lambda$-return and importance sampling.
> >
> > 2. For experiments, we conducted additional experiments to make a fairer comparison. We also added more experimental details, data statistics and discussed the computational costs in our revision.
> >
> > We sincerely hope that you consider re-evaluating our paper and raising your score if our response has addressed your concerns and issues.
> >
> > As the discussion period nears its conclusion, we would like to ask if you have any remaining concerns. We are looking forward to your feedback.
> >
> > Best regards,
> > Authors

---

> > > ### Comment · Reviewer_bDF5 · 2024-11-25
> > >
> > > I appreciate the authors' efforts to address the concerns.
> > >
> > > Regarding the additional experiments: If I understand correctly, the authors added the number of paired trajectories only for DPO to enable a fair comparison. However, the results for DQO in Tables 3 and 4 also appear to have changed for both the greedy and sampling approaches. Moreover, with the use of additional data, the performance of DPO decreases, except for Gemma-1.1-7B-it on GSM8K. This result seems counterintuitive, especially considering that the benefits of using DQO are less than 2%, as shown in both Tables 3 and 4. Additionally, based on the experimental details provided, DQO requires 2.5x the training time compared to DPO, which raises concerns about whether the improvement is worthwhile.
> > >
> > > Therefore, after carefully reviewing the response, I have decided to maintain my score.

---

> > > > ### Author Response · Authors · 2024-11-26
> > > >
> > > > Dear Reviewer bDF5
> > > >
> > > > Thank you for your feedback. We are glad that our rebuttal has effectively addressed many of your previous concerns. We would like to address your remaining concerns as follows:
> > > >
> > > > **Q1**: "However, the results for DQO in Tables 3 and 4 also appear to have changed for both the greedy and sampling approaches"
> > > >
> > > > **A1**: We would like to clarify that the experimental results have been slightly adjusted due to an update in the vLLM version. When we reran the sampling experiment, we observed some discrepancies compared to our previously reported results. These differences are attributed to an update in the roll-out package VLLM. To ensure a fairer comparison, we used the latest vLLM version (vllm 0.6.3.post1) and reran all the experiments.
> > > >
> > > > **Q2**: "Moreover, with the use of additional data, the performance of DPO decreases, except for Gemma-1.1-7B-it on GSM8K. This result seems counterintuitive."
> > > >
> > > > **A2**: Your statement that "the performance of DPO decreases, except for Gemma on GSM8K" is actually incorrect. In fact, the performance of DPO improves when additional data is used for Gemma-1.1-7B-it on both GSM8K and MATH. We present the results of DQO on Gemma-1.1-7B-it in the following table for your reference.
> > > >
> > > > |          | GSM8K (Greedy) | GSM8K (Sample) | MATH (Greedy) | MATH (Sample)  |
> > > > | -------- | -------------- | -------------- | ------------- | -------------- |
> > > > | Previous | 57.85          | 59.41          | 22.62         | 22.66          |
> > > > | Current  | 63.46          | 62.76$\pm$0.48 | 23.18         | 23.44$\pm$0.30 |
> > > >
> > > > We assume you're referring to the results of DPO on Qwen2-7B-Instruct when saying "the result seems counterintuitive". For the ease of illustration, we present the results of DQO on Qwen2-7B-Instruct as follows:
> > > >
> > > > |          | GSM8K (Greedy) | GSM8K (Sample) | MATH (Greedy) | MATH (Sample)  |
> > > > | -------- | -------------- | -------------- | ------------- | -------------- |
> > > > | Previous | 87.26          | 84.23          | 51.66         | 49.00          |
> > > > | Current  | 85.35          | 85.67$\pm$1.01 | 49.36         | 48.24$\pm$0.29 |
> > > >
> > > > We only reran the DPO experiment on MATH with additional data, as the original training dataset size for GSM8K already exceeds half of the DQO training data size. We indeed observe a performance decrease in the MATH dataset, which we believe can be explained by the over-optimization issue of DPO [1].
> > > >
> > > > **Q3**: “DQO requires 2.5x the training time compared to DPO, which raises concerns about whether the improvement is worthwhile”
> > > >
> > > > **A3**: We ackowledge that DQO requires more time than DPO. Nevertheless, as we have discussed in introduction, the key challenge in math reasoning tasks is the long horizon and sparse reward. To fully tackle this challenge, it is necessary to introduce process supervision, which will inevitably requires a larger computational cost. There is a tradeoff between computational cost and improved reasoning ability. Therefore, in scenarios where accuracy is prioritized over training time, the improvements offered by DQO are well worth the trade-off, making it especially valuable in such contexts.
> > > >
> > > > Moreover, DQO can benefit from external process value models and gain a further improvement while not introducing extra training burden. As shown in Table 7, even with the incorporation of the simplest process values, DQO outperforms DPO on average by more than 1.5% on GSM8K and 2% on MATH, under the greedy decoding setting. Therefore, we believe DQO still holds potential for further improvement, particularly if a strong process reward model becomes available.
> > > >
> > > >
> > > >
> > > > References:
> > > >
> > > > [1] Liu, Zhihan, et al. "Provably mitigating overoptimization in rlhf: Your sft loss is implicitly an adversarial regularizer." arXiv preprint arXiv:2405.16436 (2024).

---

> > > > > ### Comment · Reviewer_bDF5 · 2024-12-03
> > > > >
> > > > > Thank you for the further clarification. I now understand the differences in the experimental results. However, the marginal improvements, and the higher computational/time requirements compared to other approaches for tackling reasoning problems, lead me to maintain my scores.

---

### Official Review · Reviewer_KP7o · 2024-11-04

**Soundness:** 3
**Presentation:** 2
**Contribution:** 2
**Rating:** 3
**Confidence:** 4

**Summary:**

This paper focus on token-level optimization in LLMs. They rewrite the learning objective in LLMs and utilize the soft actor-critic (SAC) framework to optimize a Q-function directly parameterized by the language model. Experimental results on GSM8K and MATH  demonstrate the performance of proposed method.

**Strengths:**

The problem is significant, as sparse rewards cause the instability of learning process in RL.

**Weaknesses:**

It seems that the author just apply SAC in the context of LLM and rewrite the objectives.

The excessive number of formulas has made the text lengthy and difficult to follow; the paper should be written in a more concise way. Below are some suggestions.
- Eq 9 can be removed and mention to parameterize the following modules $Q_\theta$, $\pi_{\theta}$, $V_{\phi}$, as it is almost the same with Eq 8.
- Could it be more reasonable to move Eq. (6) and Eq. (7) to the preliminaries section or move them into appendix. Since Eq 6 and Eq 7 come from the previous work and it can be regarded to replace the original reward $r$ with $\bar{r}$.
- If $\pi^*$, $Q^*$ and $V^*$ are not used after the definition in Eq 3 - 5, also consider to move them into Appendix.
- Section 3.2 and Section 3.3 also can be more concise by referring to the original paper and discuss more about the challenges or special design in order to apply them in LLMs.

Additionally, minor issues in the formulas require proofreading to correct (see questions for specifics).

The experimental results over Qwen2-7B-Instruct model are not convincing:  the proposed method achieves an improvement of less than 1% on the two datasets, it is not sure if it comes from randomness. Could you please provide statistical significance tests or report the average performance/standard error over different seeds? Alternatively, other experiments over additional models or tasks to demonstrate stronger improvement?

**Questions:**

In Eq. (2), it appears $\beta$ is missing.

In Eq. (7), should it be $r$ instead of $\bar{r}$?

In Line 220, "By plugging in (9) to (7)" -> "By plugging in Eq. (9) into Eq. (7)."

If I understand correctly, there is a missing minus sign in Eq. (11) which causing the RHS of Eq. (10) and Eq. (11) sum to zero.

In Eq. (12), could you clarify the meaning of $ s_{h+1} \sim P(\cdot \mid s_h, a_h) $? Does this imply a deterministic process where $s_{h+1} = \text{Concat}(s_h, a_h) $?

In Eq. (14), should it be $ r(a_{h+l}, s_{h+l}) $?

If I’ve misunderstood any part of this, please feel free to correct me.

---

> ### Author Response · Authors · 2024-11-22
>
> Thank you for your constructive feedback! We answer your questions as follows
>
> **Q1**: It seems that the author just apply SAC in the context of LLM and rewrite the objectives.
>
> **A1**: We believe that this is a misunderstanding and would like to reiterate our contributions as follows. First, in order to overcome the sparse reward issue, instead of formulating the problem as a bandit, we consider formulating it as a MDP problem. To learn the MDP, we do not simply apply SAC but adapt it to a direct-preference-learning algorithm to eliminate the Q-function model, which can directly learn the policy. Besides, we also introduce importance sampling to tackle the distribution shift issue in offline learning and incorporate lambda-return to address the instability due to V-model initialization. Therefore, our contribution goes far beyond simply applying SAC to the context of LLM and rewriting the objectives.
>
> **Q2**: The excessive number of formulas has made the text lengthy and difficult to follow.
>
> **A2**: Thank you very much for your valuable suggestions! We have revised our paper according to your suggestions. For the second point, we think that the SAC framework, whose objective function is equations (6) and (7), is one of many approaches to learning the optimal $Q^*$ and $V^*$. Therefore, it might not be appropriate to place (6) and (7) in Section 2. In the meanwhile, the objective functions are derived based on (6) and (7), which means that deferring them to the appendix might cause the main text not self-contained.
>
> **Q3**: Additionally, minor issues in the formulas require proofreading to correct
>
> **A3**: Thank you very much for pointing out the errors! We have fixed them in our revision.
>
> **Q4**: The experimental results over Qwen2-7B-Instruct model are not convincing
>
> **A4**: Thank you for your suggestions. For sampling decoding, we sample 5 time with difference seeds and calculated the mean and standard deviation. The results are directly updated in the paper. Specifically, we summarize the updated results for Qwen here. Due to the version of vllm the results might various from previous.
>
> | Method     | Qwen  + GSM8K (greedy) | Qwen  + GSM8K (sample) | Qwen  + MATH (greedy) | Qwen  + MATH (sample) |
> | ---------- | ---------------------- | ---------------------- | --------------------- | --------------------- |
> | Base Model | 72.77                  | 60.77$\pm$1.62        | 37.44                 | 35.79$\pm$0.50        |
> | DPO        | 85.35                  | 85.67$\pm$1.01         | 49.36                 | 48.24$\pm$0.29        |
> | DRO        | 86.73                  | 82.56$\pm$0.48         | 51.84                 | 47.39$\pm$0.28        |
> | DQO        | 87.95                  | 85.13$\pm$0.47         | 51.96                 | 49.36$\pm$0.25        |
>
> The results show that DQO outperforms all other baselines significantly on MATH dataset and GSM8K greedy decoding, and comparable with DPO when doing sampling on GSM8K, which is consistant with our claims in the paper. Due to time limit, our results on other models are not availble now and we will show the result once it is available.

---

> > ### Author Response · Authors · 2024-11-25
> > **Looking Forward to Your Reply**
> >
> > Dear Reviewer KP7o:
> >
> > We hope this message finds you well. Thank you for your thoughtful and constructive feedback on our submission.
> >
> > We have answered all the comments and questions you raised. Specifically,
> >
> > 1. We correct the typos in our paper and rewrite the formulas to make them easier to follow according to your suggestions
> >
> > 2. We conducted extra experiments to show that the improvement of DQO is actually significant and not from randomness
> >
> > We sincerely hope that you consider re-evaluating our paper and raising your score if our response has addressed your concerns and issues.
> >
> > As the discussion period nears its conclusion, we would like to ask if you have any remaining concerns. We are looking forward to your feedback.
> >
> > Best regards,
> > Authors

---

> > > ### Comment · Reviewer_KP7o · 2024-11-27
> > >
> > > Thanks for your efforts in addressing my concerns. However, the results are still not convincing for me (even the maximum improvement is about 2% compared to DPO and DRO). Combined with the training cost and the method design, I would like to keep my score.

---

> > > > ### Author Response · Authors · 2024-11-28
> > > >
> > > > Thank you for your further feedback! We would like to address your remaining concerns as follows
> > > >
> > > > **Q1**: The results are still not convincing for me (even though the maximum improvement is about 2% compared to DPO and DRO)
> > > >
> > > > **A1**: We thank the reviewer for recognizing the improvement brought by our method and would like to reiterate the significance of the improvement. As demonstrated by Table 3 and Table 4, DQO outperforms DRO on all the 8 experiments and the margin surpasses 1% on 5 of totally 8 experiments. Simiarly, DQO's performance surpasses DRO on 7 of totally 8 experiments and all the improvements surpasses 1%. The results of standard deviations also show that all improvements are statistically significant. Therefore, the improvement brought by DQO is prevalent and significant.
> > > >
> > > > **Q2**: Combined with the training cost and the method design, I would like to keep my score.
> > > >
> > > > **A2**: We ackowledge that DQO are more costly than our baselines. Nevertheless, as we have discussed in introduction, the key challenge in math reasoning tasks is the long horizon and sparse reward. Our method are actually designed to fully tackle this challenge. Specifically, in these scenarios, it is necessary to introduce process supervision. This will inevitably requires a larger computational cost. There is a tradeoff between computational cost and improved reasoning ability. In scenarios where accuracy is prioritized over training time, the improvements offered by DQO are well worth the training cost, which highlights the advantage of the method design of DQO.
> > > >
> > > > We hope our answer can address your concerns. Please let us know if you have further questions.

---

### Official Review · Reviewer_Vfr4 · 2024-11-04

**Soundness:** 3
**Presentation:** 3
**Contribution:** 3
**Rating:** 3
**Confidence:** 4

**Summary:**

Direct Q-function Optimization (DQO) improves language model performance by treating text generation as a Markov Decision Process and using soft actor-critic methods to directly optimize the Q-function, demonstrating superior results on math problem-solving tasks.

**Strengths:**

* The theorem is presented in a straightforward manner.
* The experiments demonstrate promising results, particularly in the context of small-scale open-source LLMs.

**Weaknesses:**

* I believe the method can heavily rely on the accuracy of the process value. but the difficulties should be analyzed in your experiment.
* There is a lack of analysis regarding whether the process value is fairly accessible or measurable.
* The motivation appears to be aligned with the process-supervised reward model approach. Could you clarify and demonstrate the key differences between your method and theirs?

**Questions:**

* Could you provide a more in-depth analysis of your strengths when it comes to handling long-horizon tasks?
* Could you expand on how experiments can contribute to evaluating the process value? And how to produce these process values to make it could scale in practice.
* What methods or strategies can be employed to effectively generate process value?

---

> ### Author Response · Authors · 2024-11-22
>
> Thank you for your comment. We answer your questions as follows
>
> **Q1**: I believe the method can heavily rely on the accuracy of the process value. but the difficulties should be analyzed in your experiment.
>
> **A1**: This is a very good point. Indeed, thanks to our MDP formulation, our method DQO can naturally incorporate process reward and achieve even better performance. Empirically, in Section 4.4, we show that while DQO itself can already outperform all baselines, it can be further improved by integrating even the simple process rewards. Therefore, we believe that DQO can perform even better when more accurate process rewards are available. However, since the main focus of this paper is the offline reinforcement learning algorithm DQO, a detailed discussion on acquiring better process value lies out of the scope of this paper and we leave it as our future works.
>
> **Q2**: There is a lack of analysis regarding whether the process value is fairly accessible or measurable.
>
> **A2**: This is a good point. Actually, in this paper, we formulate the problem of LLM generation as a Markov Decision Process. Then, to solve this MDP, we adopt the framework of soft-actor-critic (SAC), which includes training a value function that can serve as the process value. Therefore, even if external process values are not available, the SAC framework itself already provides process supervision. Our experiment results, as shown in Table 3 and Table 4, consolidate our claim and show that DQO outperforms other baselines without process value and is thus still valuable when process value is inaccessible. As for the measurability of process value, we believe that we can construct a preference dataset and see whether the prediction of process value aligns with the ground-truth labels. However, this lies out of the scope of this paper and we leave this interesting problem as future work.
>
> **Q3**: The motivation appears to be aligned with the process-supervised reward model approach. Could you clarify and demonstrate the key differences between your method and theirs?
>
> **A3**: We believe that this is a misunderstanding. In this paper, our major contribution is an RL algorithm for LLM that does not rely on process-reward model. Specifically, we propose a DQO, which adopts the framework of soft-actor-critic and direct-preference learning, and equipped with $\lambda$-return and importance sampling to tackle the issue of inaccurate V-function initialization and offline learning. Our method indeed can further benefit from an off-the-shelf process reward models. However, as shown in Table 3 and Table 4, even without external process reward, DQO can still outperform all baseline methods.
>
> **Q4**: Could you provide a more in-depth analysis of your strengths when it comes to handling long-horizon tasks?
>
> **A4**: From a mathematical perspective, As we have demonstrated in Section 1, we show that previous works usually formulate generation tasks as a bandit problem, which makes it hard to capture the implicit long-horizon structures. Motivated by this, we formulate the task as a Markov Decision Process where each action is a token. We then adopt a soft-actor-critic framework, which includes training a value function that serves as a process-supervision. This process supervision addresses the challenge of sparse reward in long-horizon tasks and therefore benefits the training process. Empirically, as shown in Tables 3 and 4, DQO outperforms the baselines without external process supervision, and Table 7 further shows that DQO can be further boosted when stronger process-supervisions is available. This shows the DQO's advantage in dealing with such long-horizon tasks.
>
> **Q5**: Could you expand on how experiments can contribute to evaluating the process value? And how to produce these process values to make it could scale in practice.
>
> **A5**: We would like to emphasize that the primary goal of our algorithm is to output a well-performing policy. The output value is only a by-product of our approach and is not used during inference. Therefore, conducting experiments on evaluating the value model lies out of the scope of our paper and we leave it as our future work.

---

> > ### Author Response · Authors · 2024-11-22
> >
> > **Q6**: What methods or strategies can be employed to effectively generate process value?
> >
> > **A6**: This is a good question. In this paper, we adopt the most straightforward approach. As described in Section 4.4, we first split the response with a double break line into several segments. For each prefix, we prompt the language model for completion and take the best of 20. If there is a correct answer, we know that the prefix is correct and therefore we can assume that the last segment is making progress and we assign it with a positive process reward. However, the primary focus of this paper is not on generating reliable process value but on proposing an offline RL algorithm for LLM, namely DQO. When process values are available, DQO enjoys good performances, but as shown by Tables 3 and 4, even without such process value, DQO already outperforms all our baselines. We have added related comments in our revision to highlight this point. In the meanwhile, we believe that how to build is an interesting question and we leave it as our future direction.
> >
> > We appreciate the recognition of our good presentation, soundness and contribution. We sincerely hope that you can raise your score if our answers address your concerns.

---

> > > ### Author Response · Authors · 2024-11-25
> > > **Looking Forward to Your Reply**
> > >
> > > Dear Reviewer Vfr4:
> > >
> > > We hope this message finds you well. Thank you for your thoughtful and constructive feedback on our submission.
> > >
> > > We have answered all the comments and questions you raised. Specifically, we provide detailed illustrations on the relationship between our work and process-value approaches and how DQO can utilize external process value. We sincerely hope that you consider re-evaluating our paper and raising your score if our response has addressed your concerns and issues.
> > >
> > > As the discussion period nears its conclusion, we would like to ask if you have any remaining concerns. We are looking forward to your feedback.
> > >
> > > Best regards,
> > > Authors

---

> > > > ### Comment · Reviewer_Vfr4 · 2024-11-26
> > > >
> > > > Thank you for your efforts and time. I think you have addressed some of my concerns; however, I still have reservations about the scalability of the method, and I find the contribution of the theorem to be somewhat limited. I would recommend that the authors consider improving the reward design by incorporating rule-based rewards, as this could help mitigate the sparse reward problem, especially in mathematical problems where automated evaluation is possible. Consequently, while I acknowledge the improvements made, my concerns remain significant. I would have raised my score to a 4, but since there is no option for 4 and it is not sufficient for a 5, I have decided to keep my score unchanged.

---

> > > > > ### Author Response · Authors · 2024-11-26
> > > > > **We're confused about your feedback**
> > > > >
> > > > > Dear reviewer,
> > > > >
> > > > >    Thank you for your feedback. However, we are a bit confused by your comments because (1) the scalability of our method was never mentioned in your initial review, and (2) our paper does not include any theorems, nor do we claim any theorems as part of our contributions. Could it be that this feedback was mistakenly copied from a review of a different paper?
> > > > >
> > > > >    Thank you.
> > > > >
> > > > > Best,
> > > > > Authors

---

> > > > > > ### Comment · Reviewer_Vfr4 · 2024-11-26
> > > > > >
> > > > > > I mentioned scalability because I am still concerned about whether the value your model provides remains valid when applied to complex problems. While I fully understand your method, it does not seem sufficient to address challenging math problems.
> > > > > >
> > > > > > Regarding the theorem, I was referring to applying SAC in the context of LLMs. In my experience, sparse rewards present significant challenges when dealing with challenging math problems, even with a high-accuracy reward model.
> > > > > >
> > > > > > I’m sorry, but I cannot offer you a higher score at this time.

---

> > > > > > > ### Author Response · Authors · 2024-11-26
> > > > > > >
> > > > > > > Thank you for specifying your remaining concerns.
> > > > > > >
> > > > > > >
> > > > > > > >Re: I mentioned scalability because I am still concerned about whether the value your model provides remains valid when applied to complex problems. While I fully understand your method, it does not seem sufficient to address challenging math problems.
> > > > > > >
> > > > > > > We have conducted extensive experiments to demonstrate the effectiveness of our method in solving math problems from GSM8K and MATH, both of which are considered as important benchmarks in literature. Could you clarify what type of more challenging math problems you would like to see?
> > > > > > >
> > > > > > >
> > > > > > > >Re: Regarding the theorem, I was referring to applying SAC in the context of LLMs. In my experience, sparse rewards present significant challenges when dealing with challenging math problems, even with a high-accuracy reward model.
> > > > > > >
> > > > > > > Our experiments have demonstrated that our method, inspired by SAC, can effectively handle the sparse reward challenge and achieves strong results.
> > > > > > >
> > > > > > > Best,
> > > > > > > Authors

---

### Official Review · Reviewer_31Cd · 2024-11-08

**Soundness:** 2
**Presentation:** 3
**Contribution:** 2
**Rating:** 3
**Confidence:** 4

**Summary:**

The paper presents a novel offline reinforcement learning (RL) algorithm, Direct Q-function Optimization (DQO), aimed at improving the multi-step reasoning capabilities of large language models (LLMs). The authors propose formulating the response generation process as a Markov Decision Process (MDP) and utilize the soft actor-critic (SAC) framework to optimize a Q-function parameterized by the language model. The paper claims that DQO outperforms previous methods on math problem-solving datasets, GSM8K and MATH, establishing it as a promising approach for aligning language models.

**Strengths:**

1. The paper addresses a significant issue in the field of language model alignment and multi-step reasoning, which is a valuable contribution to the advancement of LLMs.
2. The presentation of the material is coherent and the paper is generally easy to follow, which aids in the understanding of the proposed DQO algorithm.

**Weaknesses:**

A primary concern is the rationality behind the derivation of the proposed method. Please see the question section below.

**Questions:**

1. The paper assumes the existence of a ground-truth reward function, which undermines the necessity of the log \pi^*(a|s) representation used later in the paper. In particular, In Lines 149-151, the authors presuppose the existence of a ground-truth reward function. Given this, why do we still need log \pi^*(a|s) as the representation of r (See Eq 8 and Eq 9)?
2. The transition from Eq(9) to Eq(11) is questionable. Eq(7) represents the value function fitting target, not the policy optimization target. It is unclear how substituting Eq(9) into Eq(7) achieves policy optimization, despite the introduction of a reward term reparameterized by \pi_\theta. The authors must elaborate on how the substitution of Eq(9) into Eq(7) facilitates policy optimization, especially considering that Eq(7) is a value function target, not a policy optimization target.
3. There is a conflict between the definitions of reward in Line 223 and Eq 14, with the former being log \pi_ref + r and the latter being \log \pi_ref/pi + r. This inconsistency needs to be resolved.
4. The rationale behind the use of importance sampling in Section 3.3 needs further justification, particularly in the context of value function learning where it is not typically necessary. Importance sampling is typically required when the sampling policy and the current policy are inconsistent during policy optimization, not during value function learning as implied by Eq 6 and Eq 7.

---

> ### Author Response · Authors · 2024-11-22
>
> Thank you for your feedback! We answer your questions as follows
>
> **Q1**: The paper assumes the existence of a ground-truth reward function, which undermines the necessity of the $\log \pi^*(a|s)$ representation used later in the paper.
>
> **A1**: We believe that this is a misunderstanding. Actually, our setup does not require a set of known rewards $r^*$. Given any unknown reward function, we can derive the optimal $Q$-function and $V$ function as in equation (4) and (5) and therefore further derive the corresponding optimal policy $\pi^*$. Since the optimal reward $r^*$ is unknown, the optimal policy $\pi^*$ is also unknown. To estimate the optimal policy $\pi^*$, we optimize the SAC objective functions defined in (6) and (7) to learn the environment. However, if we directly follow the original framework of SAC and model the Q-function and the policy separately, at inference time it requires extra combination of $Q$-value model and the policy model which is the language model (like the method in ILQL, Offline RL for Natural Language Generation with Implicit Language Q Learning), which is inefficient. Thus, inspired by direct-preference learning approach, we directly parameterize $Q$-value function by the policy $\pi$. This approach enables us to directy obtain the optimal policy $\pi^*$ if we can learn the optimal Q-function $Q^*$.
>
> **Q2**: The transition from Eq(9) to Eq(11) is questionable.
>
> **A2**: We kindly remind the reviewer that equation (7) is not value function fitting target. Instead, (7) should be viewed as a function of $\theta$ and is the Q-function fitting target. In DQO, $\theta$ is actually the parameter of the policy (i.e., the language model). Therefore, optimizing the objective (7) is indeed optimizing the policy. By substituting the $Q_\theta$ in (7) we get (11). We have add interpretion in this part in our revision to avoid further misunderstanding.
>
> **Q3**: There is a conflict between the definitions of reward in Line 223 and Eq (14)
>
> **A3**: We believe that this is a misunderstanding. Actually, as shown in equation (5), our value function is defined as $E_{\pi}[\sum_{t=h}^H \bar{r}(s_t, a_t) - \beta \sum_{t=h}^H \log \pi(a_t|s_t)]$, which is the expection of total adjusted reward $\bar{r}$ and **entropy of current policy**. Furthermore, $\bar{r}(s_t, a_t)$ can be decomposed into the **actual reward** $r(s_t, a_t)$ and the **negative entropy of the reference policy** $\log \pi_{\text{ref}}(a_t|s_t)$. Therefore, in the right hand side of (14), besides $V^{\pi}(s_{h+g})$, we also need all three terms above, that is, the actual reward $r(s_{h+l}, a_{h+l})$, the negative entropy of the reference policy $\log \pi_{\text{ref}}(a_{h+l}|s_{h+l})$ and the entropy of current policy $-\log \pi_{\theta}(a_{h+l}|s_{h+l})$. We apologize for the confusion and revised the definition of soft Q-function and V-function in our revision.
>
> **Q4**: The rationale behind the use of importance sampling in Section 3.3 needs further justification, particularly in the context of value function learning where it is not typically necessary.
>
> **A4**: This is a very good point. Actually, when applying to offline setting, in principle it is necessary to incorporate importance. Specifically, to estimate the $V$-fucntion of **current policy** $V^{\pi}(s)=E_{a\sim \pi_\theta(\cdot | s))}[Q(s,a)]$, we need to take expectation over all action $a$ with probability $\pi_\theta(a|s)$. However, since the training data is sampled from the **reference policy** $\pi_\text{ref}$, taking expectation over $\mathcal{D}$ is essentially computing $E_{a\sim \pi_{\text{ref}}(\cdot | s))}[Q(s,a)]$, which actually equals to $V^{\pi_\text{ref}}$. Therefore, to bridge the distributional mismatch here, we have to incorporate important sampling to reweight the samples to make it align with the distribution generated from $\pi_{\theta}$. This motivates the introduction of importance sampling.

---

> > ### Author Response · Authors · 2024-11-25
> > **Looking Forward to Your Reply**
> >
> > Dear Reviewer 31Cd:
> >
> > We hope this message finds you well. Thank you for your thoughtful and constructive feedback on our submission.
> >
> > We have answered all the comments and questions you raised. Specifically, we illustrated how $\pi^*$ is derived and learned, explained how equation (11) and equation (14) are derived and provided the rationale behind introducing importance sampling to DQO. We sincerely hope that you consider re-evaluating our paper and raising your score if our response has addressed your concerns and issues.
> >
> > As the discussion period nears its conclusion, we would like to ask if you have any remaining concerns. We are looking forward to your feedback.
> >
> > Best regards,
> > Authors

---

### Author Response · Authors · 2024-11-22

We sincerely thank the reviewers for handling our paper! We have uploaded the revision of our paper. Here we highlight our major revisions as follows

1. In Section 2 and Section 3, we correct the typos and add relative remarks as suggested by reviewer 31Cd, KP7o and bDF5.

2. In Section 4, as suggested by reviewer KP7o, for the sampling decoding strategy, we use different seeds and sample 5 responses for each prompt. We report the mean and standard deviation. Furthermore, we follow the suggestion of reviewer bDF5 and modified our training set of DPO for a more sounding comparison. The updated results for Qwen are summarized as follows

| Method            | GSM8K (greedy) | GSM8K (sample) | MATH (greedy) | MATH (sample) |
| ----------------- | ---------------------- | ---------------------- | --------------------- | --------------------- |
| Qwen2-7B-Instruct | 72.77                  | 60.77$\pm$1.62        | 37.44                 | 35.79$\pm$0.50        |
| SFT               | 85.06                  | 84.06$\pm$0.66         | 44.38                 | 37.43$\pm$0.41        |
| RS                | 84.15                  | 84.43$\pm$0.59         | 49.82                 | 48.19$\pm$0.43        |
| KTO               | 86.35                  | 82.56$\pm$0.48         | 50.32                 | 46.52$\pm$0.48        |
| DPO               | 85.35                  | 85.67$\pm$1.01         | 49.36                 | 48.24$\pm$0.29        |
| DRO               | 86.73                  | 82.56$\pm$0.48         | 51.84                 | 47.39$\pm$0.28        |
| DQO               | 87.95                  | 85.13$\pm$0.47         | 51.96                 | 49.36$\pm$0.25        |

The updated results for Gemma are summarized as follows

| Method          | GSM8K (greedy) | GSM8K (sample) | MATH (greedy) | MATH (sample) |
| --------------- | ----------------------- | ----------------------- | ---------------------- | ---------------------- |
| Gemma-1.1-7B-it | 39.56                   | 37.89$\pm$1.02          | 17.04                  | 16.14$\pm$0.21         |
| SFT             | 53.45                   | 46.14$\pm$1.07          | 21.64                  | 18.84$\pm$0.47         |
| RS              | 53.60                   | 53.17$\pm$0.94          | 21.74                  | 20.77$\pm$0.26         |
| KTO             | 50.49                   | 49.29$\pm$0.74          | 18.56                  | 18.58$\pm$0.17         |
| DPO             | 63.46                   | 62.76$\pm$0.48          | 23.18                  | 23.44$\pm$0.30         |
| DRO             | 62.92                   | 63.00$\pm$0.92          | 24.56                  | 24.10$\pm$0.37         |
| DQO             | 64.54                   | 64.00$\pm$0.37          | 24.90                  | 24.84$\pm$0.29         |


3. As suggested by reviewer bDF5, we add some additional details of our experiment in Appendix. Specifically, in Appendix A, we add Table 8 showing the distribution of positive and negative prompts in our collected trajectories. In appendix B, we add discussions about the computational efficiency by reporting the training time of baselines and our methods.

---

### Meta-Review · Area_Chair_1eCC · 2024-12-20

**Metareview:**

The paper tackles an important problem in language model alignment and multi-step reasoning, an area of growing significance for advancing the capabilities of Large Language Models (LLMs). Reviewers found the paper generally well-written and easy to follow, indicating a commendable clarity of presentation. However, a critical weakness was identified concerning the fundamental justification and derivation of the proposed DQO algorithm. This issue raises serious concerns about the validity and soundness of the core contribution.
While the paper possesses merits in addressing a relevant problem and presenting the material clearly, the identified weakness regarding the rationale behind the proposed method is substantial enough to warrant rejection.

**Additional Comments On Reviewer Discussion:**

While some reviewers' concerns have been addressed by the authors, the paper generally will be benefitted by another revision before acceptance.

---

### Decision · Program_Chairs · 2025-01-22

Reject